# Testing the effect of historical representations on collective identity and action

**Damilola Makanju** *, **Andrew G. Livingstone**, **Joseph Sweetman**

Department of Psychology, College of Life and Environmental Sciences, University of Exeter, Exeter, Devon, United Kingdom

☯ These authors contributed equally to this work.

* dm568@exeter.ac.uk

**Data Availability Statement:** All data and research material files are available from the Open Science Framework database (https://osf.io/esnb4/?view_only=0be6351ee8b34ea8b9543afa101e5824).

## Abstract

Historical representation of collective identity offer means of influencing the extent to which group members engage in activities in line with the collective interests of their group vs. their own individual interests. This research tested the effect of different historical representations of the African people on Africans' perceptions of African social identity and engagement in identity management strategies across two studies. In Study 1 ($N = 162$), we tested the effect of two historical representations: positive (prestigious precolonial African history and resistance to the colonial power) and negative (inhumane practices of precolonial Africans). In Study 2 ($N = 431$), we tested the effect of two historical representations: positive (prestigious precolonial African history) and negative factual (inhuman practices of precolonial Africans) while also making salient the ubiquitous historical representation of the African people (negative colonial-perspective) across all history conditions. We predicted that positive (vs. negative) historical representation would lead to more positive perceptions of African identity, which in turn would predict more collectively-oriented identity management strategies. Altogether, results provided no support for these predictions. We highlight methodological (and by extension theoretical) features–such as, psychological reactance and outgroup audience effect–which may have limited the effect of the manipulations to help inform the interpretation of the null findings obtained. We conclude by discussing other limitations and the theoretical implications of our work, before pointing out various avenues for future research to help us better test, and understand, the role of historical representation in the African context.

## Introduction

'The tragedy of Africa is that the African has not fully entered into history. The African peasant, who for thousands of years have lived according to the seasons, whose life ideal was to be in harmony with nature, only knew the eternal renewal of time, rhythmed by the endless repetition of the same gestures and the same words. In this imaginary world where everything starts over and over again there is no place for human adventure or for the idea of progress' [1] [p. 3].

**Funding:** The authors received no specific funding for this work.

**Competing interests:** The authors have declared that no competing interests exist.

'Even bigger problem is that the people of Africa and other parts of the colonized world have gone through a cultural and psychological crisis and have accepted, at least, partially, the European version of things. That means that the African himself has doubts about his capacity to transform and develop his natural environment. With such doubts, he even challenges those of his brothers who say that Africa can and will develop through the efforts of its own people' [2] [p. 21].

These two epigraphs–the former, Nicolas Sarkozy's Eurocentric historical representation of Africa and the latter, Walter Rodney's depiction of the effects such representations might have on Africans–illustrate an enormous challenge faced by many groups with a colonised past: that the legacy of colonialism fundamentally undermines a sense of a collective, meaningful, and positive history. This has considerable implications for understanding social identity [3,4], and collective behaviour [5] in such contexts. Focusing on African identity, the aim of this research is to test the effect of different historical representations on Africans' perceptions of African social identity and engagement in identity management strategies (e.g., individual vs. collective action). Our definition of African identity is a collective of individuals who identify their origin, cultural roots and/or homeland as stemming from the geographic region of the continent of Africa, and as such self-categorise themselves as being a member of the African category. In this paper, we outline the impact historical representations may have on collective identity, before considering how historical representations may affect perceptions of social identity, and the outcomes (i.e., identity management strategies) that historical representations may inspire (see Fig 1). We report two experimental studies with African samples in which we tested the effects of historical representations of African people on perceptions of African identity and identity management strategies.

## Historical representations and collective identity

History defines a path which helps to construct a group's purpose, its sense of collective identity, how it relates to other groups, and what its options are for combating present challenges [6]. As such, representations of collective history influence how group members relate to their collective identity. For instance, history is a resource that leaders (as 'identity entrepreneurs') utilise to define the essence of a group's identity and to mobilise people on the basis of that definition [7]. Historical representations thus have the potential to influence the ways in which individuals understand their group identity and act in turn to change it [8,9].

Arguably the most defining aspect of modern African history is colonialism, not least of all in shaping the current geographic borders and jurisdictions of its sovereign nation states. Furthermore, historical narratives both in the West and Africa have largely represented colonialism as good, at least on balance [2,10]. Specifically, the colonialist representation of Africa was

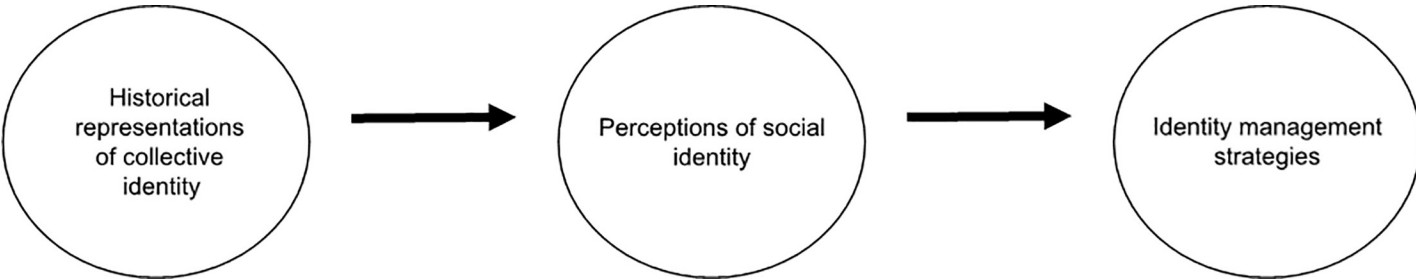

**Fig 1. The hypothesised relationship between historical representations of collective identity, perceptions of social identity, and identity management strategies.**

that the continent was 'dark', and void of civilisation [11] and that the role of Europeans was primarily as rescuers, developers, and civilisers of Africans [2]. This historical representation can negatively impact the collective self-definition and self-worth of Africans and how they shape their future [12]. In other words, the colonialist representation of Africa may undermine African identity and make it difficult for Africans to imagine and enact a positive future that utilises the present and past self-determination of Africans [13,14]. This is further exacerbated because of the perceived fit between the colonialist representation of Africa and the current 'developing' status of African countries despite gaining independence (see [15]). Indeed, anecdotal experiences of the first author in primary and secondary school education in Nigeria reflect a history curriculum void of narratives of the exploitative and oppressive nature of colonialism and of the rich, precolonial African history.

In sharp contrast to the colonialist historical narrative, precolonial African history is in fact characterized by advanced civilisations and their peoples' associated achievements [11,16]. While historical representations of Africa that resemble the colonialist narrative of a dark continent (e.g., inhumane practices by Africans) might undermine the significance attributed to African identity, precolonial African history can alternatively be represented in a way that could positively shape what it means to be African and the significance attributed to African identity. In other words, different historical representations of the African people may elicit different perceptions of one's African identity. Below, we review literature on the relationship between historical representations and perceptions of social identity and outline different ways in which social identities may be perceived by group members.

## Perceptions of social identity

Representations of a group's history should affect perceptions of social identity. For example, different historical representations held amongst group members predict levels of identification with their in-group [17,18]. Licata and colleagues [17] found, in an African sample, that regarding colonialism as exploitative positively correlated with national identification; conversely, regarding colonialism as developmental correlated negatively with national identification. Moreover, Rimé and colleagues [18] found that differences in historical representations between generations of linguistic groups in Belgium predicted French- and Dutch-speaking participants' levels of identification with Belgium and with their linguistic community. This suggests that group members may perceive their social identity differently based on the content of different historical representations.

Positive, as opposed to negative, historical representations may therefore lead to a more positive perception of one's social identity. This positive perception may help to fulfil the motivation to achieve positive distinctiveness [3,19]. This prediction is further strengthened by the notion of temporal comparisons, which posits that group members can achieve a positive social identity by comparing their group with what it used to be in the past, rather than in-the-moment comparison with other groups [20]. For example, we would expect positive historical representation of the African people that makes salient in-group ingenuity before the colonial era to lead to a more positive perception of African identity compared to a negative historical representation that makes salient inhumane practices carried out by Africans (e.g., burying alive Africans with deceased African kings). We expect this same effect when considering historical representations that make salient positive group qualities such as in-group self-determination (e.g., in narratives that depict African resistance to colonial power).

Historical representations could shape a wide variety of perceptions of social identity and related concepts. Those on which we focus in this research include: (1) the perception of one's relationship with a group—i.e., identification—such as self-definition and self-investment [21],

and collective self-esteem [22]; (2) the perception of descriptive norms–that is, what group members perceive to be normative group behaviours–associated with a social identity [23,24]; (3) the perception of the ingroup as an entity [25]; (4) the perception that a social identity might fulfil identity motives in the future [26]; and (5) the perception of the group's possible future (i.e., possible future social identities) [4].

Based on the nature of these concepts and our research context, and the argument presented above, we suggest that positive (vs. negative) historical representation will lead to positive perceptions/higher levels of all of the variables outlined above. Specifically, positive (vs. negative) historical representations will lead to (1) higher identification because of the desired aspects of social identity inherent in them [4], which should increase perceptions of the significance of relationship an individual has with the group; and (2) more positive perceptions of descriptive group norms because salience of different social categories engender varying norms that group members adhere to [27–30]. Analogously, salience of different historical representations for a broad and intra-distinctive African category [31] may engender varying perceptions of group norms based on the content of historical representations. Likewise, positive (vs. negative) representations should lead to (3) higher perceived entitativity because of the lack of common nationhood within Africa facilitated by the forceful union of African nation-states during colonialism [31,32]. Hence, positive historical representations–especially prestigious precolonial Africa–should increase perceptions of Africa as an autonomous and organic category that dates back before the advent of colonialism. Positive historical representations should also lead to (4) more positive identity motives and perceptions of the ingroup's possible future respectively, due to group members' tendencies to project the positive aspects of a group's past into the future [4].

## Historical representations and identity management strategies

The second stage of the model we test addresses how differing perceptions of social identity in turn predict behavioural outcomes. Historical narratives shape preferences for the group's future, and the identity maintenance strategies that group members adopt [4]. For example, Cinnirella's [33,34] investigation of European and national identities in Britain suggested that British national identity is generally past-oriented, partly because of Britain's former domination of world affairs and massive colonial empire. Consequently, this forms a prominent and potential barrier to Britons adopting a European identity, since Britain's past is often perceived and construed as under threat from European integration. Cinnirella [4,33,34] shows that a group's past can motivate social identity maintenance (i.e., behaviour aimed at maintaining a pre-existing, high-status social identity). Consistent with Cinnirella's work, we suggest that a group's past should also influence identity management strategies (IMS; i.e., behaviour aimed at attaining a higher-status social identity from a low-status position) [3,5]. This is because identity maintenance and management both serve the same function of assuring a high-status (positive) social identity. IMS can either be achieved individually by group members' attempts to improve their individual position and not that of the group, or collectively by group members' attempts to improve the position of the group as a whole [5].

Social identity theory (SIT) outlines three different IMS [3]. First, individuals may try to leave or dissociate themselves from their group, referred to as *individual mobility*. Individual mobility is closely related to *assimilation* [5], which occurs when a low-status group tries to emulate a higher-status group; and also *individuation*, which occurs when members of a low-status group define themselves as unique individuals and no longer as group members, and therefore can't be affected by negative group evaluations [35]. Second, individuals may try to change the favourability of intergroup comparison for the in-group, referred to as *social creativity*. This include strategies such as: *re-evaluation of comparison dimension*, which involves

devaluing the comparison dimension which defines their group's low status; *new group comparison*, which involves making comparisons with an even lower-status outgroup [19]; and *new comparison dimension*, which involves comparing the ingroup and an outgroup on new dimension(s) on which the ingroup compares more favourably, and rejecting comparisons on dimensions that will end in negative outcomes for the in-group [19]. Other social creativity strategies identified in subsequent research include *subordinate re-categorisation*, which involves improving self-evaluation by dividing the ingroup into two or more sub-groups, which are perceived to be in a higher-status position relative to that of the (superordinate) ingroup [5].

The third IMS identified in SIT is *social competition*, which involves competing directly with an outgroup to produce changes in the relative status of the ingroup. Social competition is closely related to *collective political action* which involves group members challenging a significant authority body or engaging in actions (collectively) to improve the status of the ingroup [36].

Bringing the above strands together, we suggest that positive (vs. negative) historical representations of a group may bring about positive perceptions of a social identity, which then predict IMS that aim to bring about collective improvement for the group as a whole, such as (1) engaging in social change strategies such as collective political action; and (2) adopting social creativity strategies that will improve the outcome of the group, for example, new comparison dimensions, new comparison groups, and revaluation of comparison dimension. Conversely, there should be a reduction in IMS that would *not* improve the group's outcome but rather improve the outcome of the individual, such as (1) individual mobility, individuation and assimilation; and (2) adopting social creativity strategies that will improve individual outcome, for example, subordinate re-categorisation.

## The present research

Integrating these diverse approaches to social identity, this research examines the impact of historical representations of African identity on the extent to which Africans engage in actions that benefit the collective interests of Africans (collective action) as opposed to actions that serve the personal interests of Africans (individual action). To this end, we tested the effect of presenting positive or negative historical representations of the African people, operationalised in broad terms to cover a wide spectrum of African history. We predicted that positive (vs. negative) historical representations of the African people will lead to more IMS in line with Africa's collective interest. Furthermore, we predicted that the effect of historical representations on IMS will be explained by perceptions of African identity. Specifically, positive historical representations will lead to more positive perceptions of African identity, which will, in turn, predict more IMS in line with the collective interest of Africans. To test these predictions we manipulated historical representations of the African people in two studies. These representations included positive representations such as prestigious precolonial African history and resistance to the colonial power; and negative representation such as inhumane practices of precolonial Africans.

Ethics for Study 1 (2015/954) and Study 2 (eCLESPsy000533 v8.1) was reviewed and approved by the University of Exeter, School of Psychology Ethics Review Board and all participants gave their informed consent to participate in the studies.

## Study 1: Method

### Participants

Participants were 162 African adults, of whom 47.5% were Africans living in Africa (i.e., Native Africans; N = 77), 51.2% were Africans living outside Africa (i.e., Africans in the diaspora:

N = 83) and a further 1.2% didn't report where they reside (N = 2). A sensitivity analysis using G*power 3.1 indicated that the final sample of 162 provides 80% power in the current design ($\alpha$ = .05; $df_{num}$ = 1) to detect an effect as small as Cohen's $f^2$ = 0.22 (equivalent to $\eta_p^2$ of .047) in a one-way analysis of variance (ANOVA). Our recruitment strategy was to maximise sample size given the limited time available for data collection (data were collected between the 30th of June– 1st of August of 2015). All participants were recruited using Facebook and Twitter, and an incentive was advertised whereby participants could enter a prize draw for one of two £20 ASOS vouchers. Participants were between 18 and 68 years old (M = 32.61, SD = 13.27). There were 108 males and 53 females, while one participant did not report their gender. For African nationalities, participants reported that they were: Nigerian (86.4%); Ghanaian (4.3%); Cameroonian (2.5%); Somalian (0.6%); Sierra Leonean (0.6%); African (0.6%); Nigerian and Cameroonian (0.6%); Egyptian (0.6%); Gambian (0.6%); Zimbabwean (0.6%); Ethiopian (0.6%); I don't know/not available (0.6%); and not reported (1.2%).

## Design

This study had a two-condition, between-participants design and was conducted online using the EFS survey software from Unipark. The independent variable was historical representations of the African people: positive or negative. Participants were randomly assigned to one of the two historical representation conditions. Perceptions of the African identity were conceptualised as mediator variables and identity management strategies were conceptualised as dependent variables.

## Materials

All materials can be found on the project OSF site at https://osf.io/esnb4/?view_only=0be6351ee8b34ea8b9543afa101e5824, as presented to participants. Unless otherwise indicated, responses were made on scales from 1 (*strongly disagree*) to 7 (*strongly agree*).

**Historical representations manipulation.** The materials for the historical representations manipulation were sourced from the BBC's 'lost kingdoms of Africa' documentary [11,37] and were in textual and video forms. Textually, both representations had two short paragraphs and the videos had similar durations (positive: 3:46 minutes and negative: 3:02 minutes).

*Positive history*. This representation portrayed African history in positive terms, focusing on narratives of precolonial African development, such as the architectural achievements of Great Zimbabwe in the 13th century and resistance to European colonial power, such as the Zulu victory over British troops in the battle of Isandlwana in 1879.

*Negative history*. This representation portrayed African history in negative terms, focusing on narratives of inhumane practices in precolonial Africa, such as killing and burying of African citizens with deceased Kings, and Africans selling Africans during the transatlantic slave trade.

**Manipulation check.** We checked if participants perceived the narratives as positive in the positive history condition, and negative in the negative history condition. Participants were asked the extent to which they agreed if "these aspects/details of African history are positive/negative" (depending on history condition).

**Novelty check.** A 1-item measure assessed how new the details of the historical representations were to participants. The item was: 'These aspects/details of African history are new to me'.

**Mediating variables: Perceptions of African identity.** *Group level self-definition and self-investment*. The 14-item measure ($\alpha$ = .89) of identification from Leach and colleagues [21] was used to assess participants' African identification. For self-definition, there were four

items (α = .85; e.g., 'I have a lot in common with the average African person') and for self-investment, there were 10 items (α = .86; e.g., 'I feel a bond with Africans').

*Collective self-esteem.* The 16-item measure (α = .60) from Luhtanen and Crocker [22] was used to assess participants' collective self-esteem (e.g., 'I am a worthy member of my race'). We realised that the internal consistency of this measure was low, but we still decided to proceed with it as a measure because it is a measure which has been widely validated in the literature. This rationale goes for all standardised measures (i.e., measures that have been previously used in the literature) in Study 1 which had α < .70.

*Perceived descriptive group norms.* Twenty-eight items (α = .82) were generated to assess participants' perceptions of group norms of Africans. Our operationalisation of perceived descriptive group norms is descriptive *norms* around how Africans generally evaluate Africa and the identity management strategies Africans generally employ. The overall scale was scored to signal the positivity of the descriptive norms of Africans in relation to Africa. In other words, higher scores indicated that Africans adhered to positive norms that were beneficial to Africa. Specifically, items referred to norms around identity evaluation (α = .75; 6 items; e.g., 'Africans are proud of Africa.'), individual mobility (α = .49; 5 items; e.g., 'Most Africans want to make it in the West.'), collective political action (α = .70; 5 items; e.g., 'Africans come together to solve Africa's problems.'), comparison dimensions (α = .66; 4 items; e.g., 'Most Africans value family and community more than is common in the West.') and subordinate re-categorisation (α = .61; 8 items; e.g., 'Africans see themselves only in terms of their tribal/ethnic group membership.'). The 'West' was introduced when participants were completing this measure. A definition was given in bold about the West: 'Any statement with 'West/West-ernised' throughout this survey refer to developed countries (i.e. rich and democratic countries). These countries have high standard of living and education, human rights, prosperous economies and opportunities, and so on. Examples are: United States of America, United Kingdom, Canada, France etc.'

Furthermore, we ignored the low α of the individual mobility subscale of this measure since analysis was carried out with the overall scale score which has α > .70 (the same rationale went for all subscales α < .70 which has an overall scale of α > .70).

*Identity motives.* Measures of identity motives were adapted from Vignoles, Manzi, Regalia and colleagues [38] (the distinctiveness motive was omitted). These involved participants writing down five possible futures for Africa. Participants were instructed that this could either be positive (i.e., something they hoped for) or negative (i.e., something they feared). Subsequently, participants rated each response on the identity motives of self-esteem, competence, meaning, continuity and belonging. Examples of the items (α = .98) included: 'how much would Africa being this make you feel like a competent or capable person', using a scale from 0 (*not at all true*) to 10 (*extremely true*).

**Perception of the ingroup's future: Desired and feared possible future identities.**
Based on participants written responses to the possible futures for Africa, participants also rated if these responses were desired or feared. The two items included: 'how much is this something you would like Africa to become in the future', with a response scale of 0 (*would not like this at all*) to 10 (*would like this extremely much*) and 'how much is this something you are afraid Africa will become in the future', with a response scale of 0 (*not at all afraid*) to 10 (*extremely afraid*) for desired and feared possible futures respectively.

**Contemporary problems facing Africa.**   Participants were shown contemporary problems facing Africa to give participants a springboard to respond to the identity management measures. This involved showing all participants textual and video forms of contemporary problems facing Africa, which included: water crisis, food shortage, and violence across the region. These were derived from YouTube and had varied sources, which included France 24's

[39] news coverage of violence in Central Republic of Africa, ABC's news [40] coverage of hunger crisis in West Africa, and One Global's [41] relief campaign advertisement for the water crisis in Africa.

**Dependent variables: Identity management strategies (IMS).**   Measures of IMS were adapted from Blanz and colleagues [5] unless stated otherwise. Moreover, responses were made on scales from 1 (*do not agree at all*) to 5 (*fully agree*) unless stated otherwise.

**Subordinate re-categorisation.**   Participants indicated their inclination to re-categorise at the ethnic and national level. Their African Ethnic and National group membership was derived from demographic questions at the beginning of the study and transferred to this measure. The items included: 'I consider myself more as African,' 'I consider myself more as [ethnic identity],' and 'I consider myself more as [national identity].'

**Re-evaluation of comparison dimension.**   Participants indicated their tendency to devalue the economic wealth dimension when comparing Africans and Westerners. The items included: 'Africans/Westerners consider economic wealth as', using a scale from 1 (*undesirable*) to 5 (*desirable*).

**Individual mobility.**   Four items ($\alpha$ = .88) were used to indicate participants' tendency to dissociate themselves from the African identity in order to be seen as more Western (e.g., 'I make every effort to be considered as a Westernised African').

**Assimilation.**   Four items ($\alpha$ = .71) were used to assess participants' tendency to assimilate the African identity with that of Westerners (e.g., 'We Africans should take the Westerners as a model').

**Individuation.**   Two items ($r$ = .56) were used to assess participants' tendency to define themselves as unique individuals rather than as members of a group (e.g., 'I do not consider myself as belonging to any group').

**New comparison dimension.**   Four items were used to assess participants' tendencies to increase the relevance of a new dimension relative to that of the economic situation of Africa. The dimensions included: 'economic situation,' 'social relationship,' 'quality of life,' and 'opportunities for self-actualisation'. The response scale ranged from 1 (*unimportant*) to 5 (*important*).

**New comparison group.**   Three items were used to assess participants' preference for a downward group comparison relative to the high-status Westerners group. Specifically, participants were asked 'how important is it for Africans to compare themselves with each of these following group': ' Westerners,' 'Asians,' and 'South-Americans'. The response scale ranged from 1 (*unimportant*) to 5 (*important*).

**Collective political action.**   Eight items ($\alpha$ = .87) derived from Sweetman, Spears, Livingstone, and Manstead [42] assessed participants' willingness to engage in political actions (e.g., 'sign a petition'), using a scale from 1 (*very unwilling*) to 7 (*very willing*).

## Additional measures

This study contained some additional measures whose data are not analysed here. These included a four-item scale of social dominance orientation [43], the single item measure of social identification [44] and a single item measure on participants perceptions of if the possible futures for Africa they wrote down was true of the current group [38].

## Procedure

Participants were informed that the experiment was a survey investigating Africans' opinions about Africa. This experiment had five sections. Section one involved participants completing the first portion of demographic information (e.g., the African Ethnic group, African

Nationality). In section two, participants were shown African historical representations (which varied depending on the condition) in the form of written text and video. The purpose of using these forms was to reinforce the experimental manipulation, with the text preceding the video. After participants saw the narratives, they completed the measures on the perception of African identity. Section three involved showing participants contemporary problems facing Africa in the form of written text and video, with the text preceding the video. After participants read and watched the contemporary problems, they completed the measures of identity management. Section four involved participants completing the second portion of the demographic information (i.e., gender, age, residence, the frequency of stay in Africa, and employment status).

For participants in the negative history condition, a special debrief was included in section four, to curtail any negative reactions from being shown successive negative information about Africa. This debrief included the positive historical representation of Africa that participants in the positive history condition saw but was only given in textual (bullet point) form and pointed out the fact that Africa also had contemporary successes to be proud of. Lastly, section five involved thanking and debriefing participants on the purposes, hypotheses, and the expected outcome of the research.

## Results

The bivariate correlations between all variables are reported in Table 1.

### Manipulation check

A one-sample t-test was conducted within each of the two conditions to check if the group's overall means on the respective manipulation check questions were significantly higher than the mid-point (4—*Neutral*). Results showed that for the two history conditions overall means were significantly higher than the mid-point, for positive history $t(74) = 12.63$, $p < .001$, $d = 1.46$ ($M = 5.75$, $SD = 1.20$), and negative history $t(84) = 2.82$, $p = .006$, $d = 0.31$ ($M = 4.59$, $SD = 1.92$). This reveals that on average participants in the positive and negative history condition perceived historical representations as positive and negative respectively. This indicates the aspects of history they viewed fitted our operationalisations of positive and negative history.

In order to assess if the perception of historical representations differed significantly in the positive and negative history condition, we decide to reverse score the negative history manipulation check to 1 (*negative/ less positive*) to 7 (*positive/ less negative*) and then combined it with that of the positive history manipulation check to form a new variable. This meant that both manipulation checks pointed to how positive/less negative the historical representations were, with higher scores signifying a more positive/less negative historical representation. An ANOVA was then conducted, with historical representation as the independent variable, and the newly created manipulation check variable as the dependent variable. Results showed a significant difference between perception of the historical representations as participants who saw positive history ($M = 5.75$, $SD = 1.20$) indicated that the narrative was more positive/less negative than those who saw negative history ($M = 3.41$, $SD = 1.92$), with $F(1,158) = 82.35$, $p < .001$, $\eta_p^2 = .343$. We are aware this is not an ideal approach but nonetheless conducted this analysis because of how well established the positive-negative valence dimension is in the literature.

### Novelty check

A between-participant ANOVA with historical representation as a two-level factor revealed a significant effect of historical representation on how new participants perceived aspects of African history they watched, $F(1, 157) = 30.99$, $p < .001$, $\eta_p^2 = .165$, with participants in the

**Table 1.  Bivariate correlations for all variables (i.e., identity management strategies and perceptions of African identity).**

| | 1 | 2 | 3 | 4 | 5 | 6 | 7 | 8 | 9 | 10 | 11 | 12 | 13 | 14 | 15 | 16 | 17 | 18 |
|---|---|---|---|---|---|---|---|---|---|---|---|---|---|---|---|---|---|---|
| **1. Mob** | 1 | | | | | | | | | | | | | | | | | |
| **2. Ass** | .67** | 1 | | | | | | | | | | | | | | | | |
| **3. Indv** | .16* | .28** | 1 | | | | | | | | | | | | | | | |
| **4. Nat R** | -.09 | -.13 | -.12 | 1 | | | | | | | | | | | | | | |
| **5. Eth R** | -.35** | -.18* | -.11 | .36** | 1 | | | | | | | | | | | | | |
| **6. Re-eva** | .02 | .16* | .06 | .00 | .00 | 1 | | | | | | | | | | | | |
| **7. Soc R** | .05 | .03 | -.09 | .01 | .04 | .05 | 1 | | | | | | | | | | | |
| **8. QOL** | -.03 | .00 | .08 | -.09 | .05 | .02 | .39** | 1 | | | | | | | | | | |
| **9. OFS** | -.04 | -.00 | -.04 | .14 | .11 | .07 | .38** | .51** | 1 | | | | | | | | | |
| **10. AS** | -.17* | -.16 | -.12 | .11 | .06 | -.10 | -.11 | -.07 | .01 | 1 | | | | | | | | |
| **11. SA** | -.14 | -.11 | -.20* | .17* | .10 | -.05 | -.05 | -.02 | .00 | .65** | 1 | | | | | | | |
| **12. Colact** | -.16* | -.12 | -.17* | .18* | .16* | .08 | .02 | -.05 | .03 | .11 | .21** | 1 | | | | | | |
| **13. Self-D** | -.09 | -.15 | -.37** | .19* | .10 | .14 | .13 | .02 | .08 | .01 | .16* | .29** | 1 | | | | | |
| **14. Self-I** | .01 | .06 | -.27** | .18* | .07 | .14 | .17* | .08 | .08 | -.05 | .11 | .15 | .65** | 1 | | | | |
| **15. CSE** | -.15 | -.08 | -.45** | .19* | .16 | .02 | .12 | -.01 | .07 | .02 | .14 | .18* | .61** | .40** | 1 | | | |
| **16. PGN** | -.09 | -.06 | -.04 | -.12 | .17* | .17* | -.01 | .08 | .00 | -.06 | .04 | .09 | .38** | .25** | .29** | 1 | | |
| **17. IDM** | .07 | .14 | -.24** | -.03 | .09 | .05 | .17* | .04 | .11 | -.01 | .07 | .16 | .31** | .31** | .30** | .23** | 1 | |
| **18. DAF** | -.11 | -.09 | -.13 | -.05 | .07 | -.02 | .03 | -.04 | .11 | .15 | .19* | .19* | .14 | .21* | .13 | .27** | .49** | 1 |
| **19. FAF** | .23** | .19* | .10 | .11 | .01 | -.02 | .12 | .16 | .10 | -.20* | -.22** | .04 | -.02 | .10 | .05 | -.01 | -.10 | -.51** |

*p < .05.

**p < .01.

1. Mob, individual mobility; 2. Ass, assimilation; 3. Individuation; 4. Nat R, national subordinate re-categorisation; 5. Eth R, ethnic subordinate re-categorisation; 6. Re-eva, re-evaluation of comparison dimension; 7. Soc R, social relationships new comparison dimension; 8.QOL, quality of life new comparison dimension; 9. OFS, opportunities for self-actualisation new comparison dimension; 10. AS, Asians new comparison group; 11. SA, South Americans new comparison group; 12. Colact, political collective action; 13. Self-D, self-definition; 14. Self-I, self-investment; 15. CSE, collective self-esteem; 16. PGN, perceived group norms; 17. IDM, identity motives; 18. DAF, desired possible African futures; 19. FAF, feared possible African futures.

positive history condition ($M$ = 4.31, $SD$ = 2.08) perceiving the aspects of history as more novel in comparison to participants in the negative history condition ($M$ = 2.63, $SD$ = 1.71).

## Perception of African Identity

A between-participant MANOVA with historical representation as a two-level factor was conducted to test the effect of historical representation on perceptions of the African identity. Using Wilks' lambda, there was a non-significant effect of historical representation on perceptions of African identity, $\lambda$ = 0.94, $F(7, 136)$ = 1.32, $p$ = .246, $\eta_p^2$ = .064. Moreover, results from the follow-up analyses from the univariate ANOVAs within the MANOVA revealed consistent null effects of the historical representation manipulation on all variables ($F$s < 1.73, $p$s > .191, $\eta_p^2$s < .012). We report all relevant inferential and descriptive statistics of all ANOVAs in Tables 2 and 3 respectively.

## Identity management strategies

A similar MANOVA was conducted to test the effect of historical representation on identity management strategies. Using Wilks' lambda, there was a non-significant effect of historical representation on identity management strategies, $\lambda$ = 0.94, $F(12, 134)$ = 0.73, $p$ = .720, $\eta_p^2$ = .061. Moreover, results from the follow-up analyses from the univariate ANOVAs within the

**Table 2. ANOVA results for the effect of historical condition on the perceptions of African identity.**

| Perceptions of African identity | F | df | P | $\eta_p^2$ |
|---|---|---|---|---|
| Self-definition | 0.19 | 1,142 | .667 | .001 |
| Self-investment | 0.94 | 1,142 | .334 | .007 |
| Collective self-esteem | 1.58 | 1,142 | .211 | .011 |
| Perceived descriptive group norms | 1.73 | 1,142 | .191 | .012 |
| Identity motives | 0.45 | 1,142 | .503 | .003 |
| Desired possible future African identities | 0.03 | 1,142 | .864 | .000 |
| Feared possible future African identities | 0.70 | 1,142 | .403 | .005 |

MANOVA revealed consistent null effects of historical representation on all identity management strategies ($Fs < 5.11$, $ps > .067$, $\eta_p^2 s < .023$). We report all relevant inferential and descriptive statistics of all ANOVAs in Tables 4 and 5 respectively. Due to the lack of an effect of the historical representation manipulation on all identity management strategies we did not go ahead with tests of the indirect effect of historical representations on identity management strategies via perception of African identity.

## Discussion

The primary aim of this study was to examine the effect of (positive and negative) historical representations of African people on Africans' perceptions of African identity and identity management strategies. We predicted that a positive historical representation compared to negative historical representation would lead to more positive perceptions of African identity, which would, in turn, predict more collectively-oriented identity management strategies. Altogether, results provided no support for our predictions: the historical representations manipulation had no impact on perceptions of the African identity or on identity management strategies.

We reasoned that the null effects could have arisen because the historical representation manipulation–especially the positive history condition–did not explicitly challenge the ubiquitous colonialist narrative of Africa which participants might have been used to (as indicated by the higher novelty score in the positive representation condition), and held before and during the study. In an attempt to address this limitation, Study 2 aimed to (1) make salient the colonialist narrative of Africa by depicting it side-by-side with alternative representations of African history (e.g., prestigious precolonial Africa) as a means of emphasising the contrast; (2) strengthen the positive historical representation manipulation by adopting a narrative that explicitly rebuts the colonialist narrative of Africa; and (3) restricting the narratives of the positive historical representation to the precolonial era due to this being history before the disruption from colonialism.

**Table 3. Means and standard deviations for perceptions of African identity based on historical conditions.**

| Perceptions of African identity | Positive | | Negative | |
|---|---|---|---|---|
| | M | SD | M | SD |
| Self-definition | 5.71 | 0.96 | 5.77 | 0.83 |
| Self-investment | 5.08 | 1.26 | 5.27 | 1.13 |
| Collective self-esteem | 5.15 | 0.70 | 5.01 | 0.70 |
| Perceived descriptive group norms | 3.95 | 0.54 | 3.83 | 0.56 |
| Identity motives | 7.07 | 2.72 | 7.37 | 2.59 |
| Desired possible future African identities | 8.52 | 2.45 | 8.44 | 2.66 |
| Feared possible future African identities | 5.27 | 2.66 | 4.89 | 2.69 |

**Table 4. ANOVA results for the effect of historical condition on identity management strategies.**

| IMS | F | Df | P | $\eta_p^2$ |
|---|---|---|---|---|
| Individual mobility | 0.46 | 1,145 | .487 | .003 |
| Assimilation | 1.04 | 1,145 | .237 | .010 |
| Individuation | 0.92 | 1,145 | .350 | .006 |
| National subordinate re-categorisation | 0.12 | 1,145 | .757 | .001 |
| Ethnic subordinate re-categorisation | 0.41 | 1,145 | .611 | .002 |
| Revaluation comparison dimension | 1.83 | 1,145 | .208 | .011 |
| New comparison dimension: social relationships | 0.16 | 1,145 | .570 | .002 |
| New comparison dimension: quality of life | 0.05 | 1,145 | .647 | .001 |
| New comparison dimension: opportunity of self-actualisation | 0.35 | 1,145 | .421 | .004 |
| New comparison group: Asians | 0.92 | 1,145 | .374 | .005 |
| New comparison group: South Americans | 0.11 | 1,145 | .749 | .001 |
| Collective political action | 5.11 | 1,145 | .067 | .023 |

## Study 2

We aimed to improve our approach by making a number of methodological changes from Study 1, which included: (1) the inclusion of a reflection item after the manipulation which asked participants to write down their thoughts and feelings to the historical representations they viewed, in order to strengthen participants engagement and processing of the material in the manipulation; (2) the improvement of the manipulation check by adopting the same semantic differential measures across treatment conditions to be sure that participants who, for example, view our operationalisation of negative history perceive it as more negative relative to participants who view the positive history and vice versa; (3) the removal of information on the contemporary issues facing Africa due to the potential of it overshadowing the effect of the positive historical representation on participants' identity management; (4) the inclusion of perceived group entitativity (i.e., the perceived 'realness' of African identity) [25] as an additional dimension of perception of African identity; (5) the inclusion of a measure of social competition in place of the new comparison group measure. This change was made because social competition more directly addresses the willingness to act collectively for the group [3]; and (6) the introduction of a baseline–that is, negative colonial-perspective–historical

**Table 5. Means and standard deviations for identity management strategies (IMS) based on historical conditions.**

| IMS | Positive | | Negative | |
|---|---|---|---|---|
| | M | SD | M | SD |
| Individual mobility | 2.45 | 0.94 | 2.57 | 1.01 |
| Assimilation | 2.51 | 0.91 | 2.68 | 0.81 |
| Individuation | 2.54 | 1.06 | 2.70 | 0.99 |
| National subordinate re-categorisation | -0.20 | 1.09 | -0.14 | 1.14 |
| Ethnic subordinate re-categorisation | 0.27 | 1.14 | 0.38 | 1.34 |
| Revaluation comparison dimension | -0.17 | 1.01 | 0,05 | 1.12 |
| New comparison dimension: social relationships | -0.13 | 0.68 | -0.19 | 0.73 |
| New comparison dimension: quality of life | 0.10 | 0.52 | 0.06 | 0.41 |
| New comparison dimension: opportunity of self-actualisation | -0.07 | 0.77 | -0.17 | 0.70 |
| New comparison group: Asians | -0.26 | 1.51 | -0.42 | 1.01 |
| New comparison group: South Americans | -0.39 | 1.17 | -0.44 | 0.94 |
| Collective political action | 5.08 | 1.41 | 5.46 | 1.03 |

representation in order to enhance the contrast with other historical representations. In summary, Study 2 tested the effect of two forms of historical representations, namely positive (prestigious precolonial Africa) and negative factual (inhumane practices of precolonial Africans) on perceptions of African identity and identity management strategies of Africans.

## Method

### Participants

Participants were 431 African adults, of whom 34.8% were Africans living in Africa (i.e., Native Africans; N = 150), 62.9% were Africans living outside Africa (i.e., Africans in the diaspora; N = 271) and a further 2.3% didn't report where they reside (N = 10). A sensitivity analysis using G*power 3.1 indicated that the final sample of 431 provides 80% power ($\alpha$ = .05; $df_{num}$ = 2) to detect an omnibus effect as small as Cohen's $f^2$ = 0.15 (equivalent to $\eta_p^2$ of .022) in a one-way analysis of variance (ANOVA). Our recruitment strategy was to maximise sample size given the available funds through Departmental allocation (i.e., we had a budget and recruited until it was exhausted). Initially, this study was advertised via Facebook, with a direct link to the study in an advert. This advert included an incentive whereby participants could enter a prize draw for one of five £30 ASOS vouchers. However, only 25 participants completed the study via Facebook advert. The study link was then circulated on social media platforms (e.g., Facebook and WhatsApp). A further 206 participants completed the study via social media platforms. Furthermore, 200 participants were recruited via Prolific Academic and were paid a total of £2.50 each for their participation. All participants were aged 18 to 75 (M = 33.02, SD = 13.48).

There were 237 males and 185 females, while one identified their gender as 'other' and two preferred to 'rather not say' (a further six did not report their gender). For African nationalities, participants reported that they were: Nigerian (61%); Ghanaian (6.3%); Kenyan (3.5%); African American (2.3%); Zimbabwean (1.6%); Congolese (1.4%); Zambian (1.2%); Cameroonian (1.2%); Angolan (0.9%); Ugandan (0.9%); Sierra Leonean (0.7%); Ethiopian (0.7%); Sudanese (0.7%); South African (0.7%); Tanzanian (0.7%); Malawian (0.7%); Somalian (0.5%); Soa Tomean (0.5%); Moroccan (0.5%); Togolese (0.5%); West African (0.5%); African (0.5%); Gambian (0.5%); Botswanan (0.5%); Sudanese (0.5%); Togolese (0.5%); Burundian (0.2%); Ethiopian & Ivorian (0.2%); Botswanan (0.2%); Nigerian & Ugandan (0.2%); Tunisian (0.2%); Kemet (0.2%); Libyan (0.2%); Shabazz (0.2%); Motswanan (0.2%); Creole (0.2%); Liberian (0.2%); Senegalese (0.2%); Mozambican (0.2); I don't know/not available (6%); and not reported (1.6%). Moreover, 71.7% of participants indicated that they had African citizenship and 23.7% of participants indicated that they had dual citizenship (i.e., they were concurrently citizens of an African state and another state).

### Design

This study had a three-condition, between-participants design and was conducted online using the EFS survey software from Unipark. The study's independent variable was historical representations of the African people and had three levels: negative colonial-perspective (which was conceptualised as a baseline–meaning that every participant watched this historical representation before moving on to answer the questionnaire or to watch a contrasting historical representation); positive; and negative factual. In other words, participants were randomly assigned to one of the three conditions: negative colonial-perspective condition (only); negative colonial-perspective + positive condition; and negative colonial-perspective + negative factual condition. Perceptions of African identity were conceptualised as mediator variables and identity management strategies were conceptualised as dependent variables.

## Materials

All materials can be found on the project OSF site at https://osf.io/esnb4/?view_only=0be6351ee8b34ea8b9543afa101e5824, as presented to participants. Unless otherwise indicated, responses were made on scales from 1 (*strongly disagree*) to 7 (*strongly agree*).

## Historical representations manipulation

The historical representations were derived from different documentary sources which include: Racism: a history [45], Africa's great civilisations [16], Lost kingdoms of Africa [37], Huntley Archive's [46] video on Colonial Africa and Kaci's [47] video on African Pygmy thrills by Eugene W. Castle. In terms of the content of the representations, all conditions covered three topic areas of African history. These areas were introduced using subheadings which meant that each condition of African history had three subheadings except for the negative factual condition. Specifically, in the negative factual condition, the last two aspects of African history were covered under one subheading because they were inextricably intertwined. In addition, each condition (i.e., video narratives) started with the same picture which had the lettering 'The History of Africa'. Moreover, the duration of the video narratives in minutes were 5.59 for negative colonial-perspective, 4.49 for positive, and 5.00 for negative factual.

**Negative colonial-perspective history.**   The negative colonial-perspective historical representation portrayed African history in negative terms and was intended to represent the colonialist view of Africa which was a justification of why Africa needed to be colonised by the Europeans. The subheadings of this condition were: *animalistic*, *uncivilised*, and *necessity of colonialism*.

**Positive history.**   The positive historical representation portrayed African history in positive terms and was intended to represent a decolonised version of precolonial African history by presenting high achievements of Africans which showed a highly civilised people before the advent of colonialism. The subheadings of this video were: *great scholarship*, *civilised people* and *stunning artistry*.

**Negative factual history.**   The negative factual historical representation portrayed African history in negative, but factual terms (as opposed to the non-veridical content of the negative colonial-perspective condition), focusing on inhuman practices of precolonial Africans. The subheadings of this video were: *human sacrifices* and *loss of skilled Africans and greed in African-led slave trade*.

## Reflection on manipulations

After watching the historical representations, participants were asked to reflect on the video in order to increase engagement with the historical narratives before completing the rest of the questionnaire (however, participants in the negative colonial-perspective only condition were not asked this reflection question and every other check question). Participants were asked: 'What are your thoughts and feelings on the history of Africa based on the video narratives you have seen.' Participants were allowed a maximum of 150 words in response to this question, and were instructed: 'Please don't spend too much time on this question, just write your main point and move on to answer the other questions'.

## Manipulation check

A scale to assess the valence of the aspects of African history in the second video in the positive and negative factual treatment conditions consisted of three semantic differential items ($\alpha$ = .94). These included: 'good-bad', 'pleasant-unpleasant' and 'positive-negative'. The items were prefaced with the statement 'The aspects/details of African history in the second video clip

are'. Responses were scored from 1 (*positively anchored scale end*) to 7(*negatively anchored scale end*). However, the response scale as visible to participants was not numbered to avoid attaching implied value to one type of response.

## Novelty check

We employed the same 1-item as in Study 1.

## Mediating variables: Perceptions of African identity

**Group level self-definition and self-investment.**   We employed the same measures of group level self-definition ($\alpha$ = .85) and self-investment as in Study 1 ($\alpha$ = .87).

**Collective self-esteem.**   We employed the same measures ($\alpha$ = .80) as in Study 1.

**Perceived descriptive group norms.**   Eleven items ($\alpha$ = .84) assessed participants' perception of group norms of Africans. We decided to narrow down the perceived descriptive group norms to that of identity evaluation, collective action and individual mobility due to the large number of items and lack of internal consistency within subscales in Study 1. Moreover, these items were selected and subjected to factor analysis from responses of Study 1 to confirm a 3-factor solution for the respective norm content.

Items referred to descriptive norms around identity evaluation ($\alpha$ = .83; 4 items; e.g., 'Africans are proud of Africa'), collective action tendencies ($\alpha$ = .75; 3 items; e.g., 'Africans easily come together to solve Africa's problems') and individual mobility tendencies ($\alpha$ = .61; 3 items; e.g., 'Most Africans want to it in the West'). This measure was treated in the same way as in Study 1. A definition of the 'West' was provided to participants when completing this measure. Specifically, participants were instructed: 'The word 'West' or 'Westernised' or 'Westerners' in this questionnaire refers to a set of countries (e.g., United States of America, France, United Kingdom, Russia, Germany, Canada, Australia etc.) who are wealthier and whose citizens enjoy a higher standard of living than the rest of the world'. Moreover, we ignored the low $\alpha$ of the individual mobility subscale because it was the score of the overall scale that was used for analysis which has $\alpha$ > .70.

**Group entitativity.**   The 6-item measure ($\alpha$ = .78) from Newheiser and colleagues [25] was adapted to assess participants' perception of Africa as an entity (e.g., 'Africans frequently interact with one another').

**Identity motives.**   The 6-item ($\alpha$ = .94) measure was adapted to assess participants identity motives [38]. These involved participants writing down one possible future for Africa, and participants were instructed that this could either be something they hoped for or something they feared. Subsequently, participants rated their response on the six identity motives (e.g., 'If what I wrote above became true this would give me a sense of self-esteem—the feeling that you are a likeable and worthwhile person'), using a scale from 1 (*not at all true*) to 7 (*extremely*).

**Desired and feared possible African futures.**   These measures were also derived from participants' responses to the possible future for Africa. Participants rated if their response were desired or feared. The items included: Thinking about what you wrote above, 'how much is this something you would like Africa to become in the future', with a response scale from 1 (*would not like this at all*) to 7 (*would like this extremely much*), and 'how much is this something you are afraid Africa will become in the future', with a response scale from 1 (*not at all afraid*) to 7 (*extremely afraid*) for desired and feared possible African futures respectively.

## Dependent variables: Identity management strategies (IMS)

Measures of IMS were adapted from Study 1 unless stated otherwise.

**Individual mobility, assimilation, and individuation.**   We employed the same measure for individual mobility ($\alpha$ = .93), assimilation, ($\alpha$ = .77), and individuation ($r$ = .60) as in Study 1.

**Subordinate re-categorisation.**   Participants indicated their tendency to re-categorise at National and Ethnic levels. Their African National and Ethnic group membership was derived from demographic questions at the beginning of the study and transferred to this measure. The items included: 'To what degree do you regard yourself as National identity', 'To what degree do you regard yourself as Ethnic identity', and 'To what degree do you regard yourself as an African'.

**Social competition.**   Three items ($\alpha$ = .80) were used to assess participants' beliefs about the extent to which Africa can compete with the West (e.g., 'It is our goal as Africans to not be always taught by the West, but to teach them also').

**Revaluation of comparison dimension.**   We employed the same measure of revaluation of comparison dimension as in Study 1.

**New comparison dimension.**   Eight items were used to assess participants' tendency to increase the importance of a new dimension (e.g., 'happiness') relative to that of the economic situation of Africa, using a scale from 1 (*not at all important*) to 7 (*extremely important*).

**Collective political action.**   Ten items ($\alpha$ = .90) adapted after items from Sweetman and colleagues [42] were used to assess participants' willingness to engage in political actions (e.g., 'help organise a petition'), using a scale from 1 (*very unwilling*) to 7 (*very willing*).

## Additional measures

This study contained some additional measures whose data are not analysed here. These were 3-item scales of social-structural factors (adapted from Mummendey and colleagues [48]) which included: stability, legitimacy and permeability of group boundaries, and a single item measure on participants perceptions of if the possible future for Africa they wrote down was true of the current group (adapted from Vignoles, Manzi, Regalia, et al., [38]).

## Procedure

Participants were informed that the experiment was a survey containing video representations of African history and questionnaires on their opinion of Africa. This experiment had five sections. Section one involved participants completing the first portion of demographic information (i.e., African Nationality and African Ethnicity). In section two, participants were shown African historical narratives. Specifically, all participants viewed the negative colonial-perspective historical representation first because it was conceptualised as the baseline history of Africa–which is history that is generally known and held to be legitimate by Africans and represents the colonialist accounts of precolonial African history. Some participants only watched the baseline history video (negative colonial-perspective history condition), whilst others in addition (to the baseline history) either watched the positive history video or the negative factual history video. This approach was adopted in order to give participants a platform for comparison between the negative colonial-perceptive of Africa and other historical representations. In other words, the negative colonial-perspective historical representation facilitated comparison that would heighten the impact of the second video in the other conditions.

After participants saw the historical representations, they completed the measures on the perception of African identity. Section three involved participants completing measures on identity management strategies. Section four involved participants completing the second portion of the demographic information (i.e., citizenship, gender, age, education level, residency and time spent in Africa). For participants in the negative colonial-perspective and negative

factual history conditions, an additional debrief was included in section four, to curtail any negative reactions from being shown negative information about Africa. This debrief clarified the conceptual stance of the negative historical representations that participants watched and gave them the historical representations that participants watched in the positive history condition in textual form (i.e., bullet points). Lastly, section five involved thanking and debriefing participants on the purposes, hypotheses, and the expected outcome of the research.

## Results

The bivariate correlations between all variables are reported in Table 6

### Manipulation check

A between-participant ANOVA with historical representation as a three-level factor revealed a significant effect of historical representation on how participants perceived the valence of aspects of African history in the treatment condition videos, $F(1, 259) = 351.13$, $p < .001$, $\eta_p^2 = .576$, with participants in the positive historical representation condition ($M = 1.63$, $SD = 1.00$) perceiving the aspects of history as more positive in comparison to participants in the negative historical representation condition ($M = 4.87$, $SD = 1.70$).

### Novelty check

A between-participant ANOVA with historical representation as a three-level factor revealed a marginally significant effect of treatment condition on how new participants perceived aspects of African history they watched, $F(1, 261) = 3.78$, $p = .053$, $\eta_p^2 = .014$, with participants in the positive history condition ($M = 3.21$, $SD = 2.01$) perceiving the aspects of history as more novel in comparison to participants in the negative factual history condition ($M = 2.77$, $SD = 1.66$).

### Perception of African identity

A between-participant MANOVA with historical representation as a three-level factor was conducted to test the effect of historical representation on perceptions of African identity. Using Wilks' lambda, there was a non-significant effect of historical representation, $\lambda = 0.97$, $F(16, 818) = 0.83$, $p = .656$, $\eta_p^2 = .016$. Moreover, results from the follow-up analyses from the ANOVAs revealed consistent null effects of the historical representation manipulation on all variables ($Fs < 1.56$, $ps > .211$, $\eta_p^2s < .007$). We report all relevant inferential and descriptive statistics of all ANOVAs in Tables 7 and 8 respectively.

### Identity management strategies

A similar MANOVA was conducted to test the effect of historical representation on identity management strategies. Using Wilks' lambda, there was a non-significant effect of historical representation on identity management strategies, $\lambda = 0.94$, $F(30, 776) = 0.87$, $p = .664$, $\eta_p^2 = .033$. Moreover, results from the follow-up analyses from the ANOVAs revealed consistent null effects of historical representation on all identity management strategies ($Fs < 1.64$, $ps > .195$, $\eta_p^2s < .008$). We report all relevant inferential and descriptive statistics of all ANOVAs in Tables 9 and 10 respectively. Due to the lack of an omnibus effect of the historical representation manipulation on all identity management strategies we decided not to go ahead with tests of the indirect effect of condition on identity management strategies via perceptions of African identity.

**Table 6. Bivariate for all variables (i.e., identity management strategies and perceptions of African identity).**

| | 1 | 2 | 3 | 4 | 5 | 6 | 7 | 8 | 9 | 10 | 11 | 12 | 13 | 14 | 15 | 16 | 17 | 18 | 19 | 20 | 21 | 22 |
|---|---|---|---|---|---|---|---|---|---|---|---|---|---|---|---|---|---|---|---|---|---|---|
| **1. Mob** | 1 | | | | | | | | | | | | | | | | | | | | | |
| **2. Ass** | .55** | 1 | | | | | | | | | | | | | | | | | | | | |
| **3. Indv** | .33** | .26** | 1 | | | | | | | | | | | | | | | | | | | |
| **4. Soc C** | -.02 | -.06 | -.06 | 1 | | | | | | | | | | | | | | | | | | |
| **5. Nat R** | -.02 | -.01 | -.09 | -.04 | 1 | | | | | | | | | | | | | | | | | |
| **6. Eth R** | .04 | -.06 | -.03 | -.03 | .64** | 1 | | | | | | | | | | | | | | | | |
| **7. Re-eva** | .06 | .07 | .06 | .03 | -.05 | .02 | 1 | | | | | | | | | | | | | | | |
| **8. Edu** | .02 | -.10* | -.04 | .01 | .00 | .06 | .06 | 1 | | | | | | | | | | | | | | |
| **9. Hap** | .01 | -.03 | -.05 | -.05 | .08 | .03 | .13** | .53** | 1 | | | | | | | | | | | | | |
| **10. Cul** | .01 | -.04 | -.09 | -.03 | .08 | .10* | .07 | .44** | .65** | 1 | | | | | | | | | | | | |
| **11. Com** | -.00 | -.05 | -.10* | -.02 | .10* | .10 | .03 | .48** | .74** | .77** | 1 | | | | | | | | | | | |
| **12. Rel** | .01 | .05 | -.06 | -.05 | .05 | .07 | .05 | .39** | .53** | .59** | .61** | 1 | | | | | | | | | | |
| **13. Mor** | -.02 | -.10* | -.09 | .05 | .12* | .10* | .04 | .44** | .65** | .75** | .78** | .62** | 1 | | | | | | | | | |
| **14. Fam** | -.05 | -.07 | -.08 | .01 | .12* | .11* | .05 | .44** | .68** | .76** | .79** | .69** | .83** | 1 | | | | | | | | |
| **15. Colact** | -.02 | -.07 | -.25** | .26** | .04 | -.01 | .04 | -.00 | -.10 | -.04 | -.03 | -.12* | .01 | -.09 | 1 | | | | | | | |
| **16. Self-D** | -.06 | .02 | -.26** | .14** | .13** | -.01 | -.04 | -.00 | .03 | .01 | .07 | -.02 | .04 | -.02 | .23** | 1 | | | | | | |
| **17. Self-I** | -.19** | -.18** | -.43** | .27** | .18** | .08 | -.06 | .06 | .01 | .02 | .09 | -.01 | .09 | .02 | .40** | .59** | 1 | | | | | |
| **18. CSE** | -.18** | -.18** | -.46** | .24** | .05 | -.04 | -.06 | .04 | .03 | .03 | .10* | .00 | .07 | .05 | .33** | .38** | .56** | 1 | | | | |
| **19. PGN** | -.01 | -.06 | -.00 | .13** | .03 | .03 | .05 | .03 | -.04 | -.00 | .02 | -.13** | -.05 | -.08 | .11** | .16** | .18** | .20** | 1 | | | |
| **20. Ent** | -.02 | -.03 | -.17** | .36** | .06 | .03 | .14** | .02 | -.01 | .04 | .04 | -.01 | .05 | -.01 | .26** | .48** | .36** | .32** | .37** | 1 | | |
| **21. IDM** | .17** | .13** | -.18** | .21** | .16** | .01 | .08 | -.03 | -.01 | -.08 | -.04 | -.03 | -.04 | -.07 | .25** | .35** | .37** | .34** | .18** | .33** | 1 | |
| **22. DAF** | .01 | -.05 | -.08 | .21** | .06 | .12* | .02 | .01 | -.00 | -.02 | .03 | .03 | .08 | .07 | .12* | .11* | .12* | .10* | .10* | .10* | .33** | 1 |
| **23. FAF** | .06 | .14** | -.02 | -.16** | .05 | -.09 | -.05 | -.04 | -.03 | .04 | -.02 | .00 | -.01 | -.03 | .02 | .03 | -.02 | -.02 | -.16** | -.01 | -.03 | -.50** |

*p < .05.

**p < .01.

1. Mob, individual mobility; 2. Ass, assimilation; 3. Individuation; 4. Soc C, social competition; 5. Nat R, national subordinate re-categorisation; 6. Eth R, ethnic subordinate re-categorisation; 7. Re-eva, re-evaluation of comparison dimension; 8. Edu, education new comparison dimension; 9.Hap, happiness new comparison dimension; 10. Cul, culture new comparison dimension; 11. Com, community harmony new comparison dimension; 12. Rel, religion new comparison dimension; 13. Mor, social morals new comparison dimension; 14. Fam, family new comparison dimension; 15. Colact, political collective action; 16. Self-D, self-definition; 17. Self-I, self-investment; 18. CSE, collective self-esteem; 19. PGN, perceived group norms; 20. Ent, group entitativity 21. IDM, identity motives; 22. DAF, desired possible African futures; 23. FAF, feared possible African futures.

**Table 7. ANOVA results for the effect of historical condition on the perceptions of African identity.**

| Perceptions of African identity | F | df | p | $\eta_p^2$ |
|---|---|---|---|---|
| Self-definition | 1.08 | 2. 415 | .340 | .005 |
| Self-investment | 0.65 | 2, 415 | .522 | .003 |
| Collective self-esteem | 1.56 | 2, 415 | .211 | .007 |
| Perceived descriptive group norms | 1.06 | 2. 415 | .348 | .005 |
| Group entitativity | 0.18 | 2, 415 | .833 | .001 |
| Identity motives | 0.73 | 2, 415 | .483 | .003 |
| Desired possible future African identities | 1.16 | 2, 415 | .313 | .006 |
| Feared possible future African identities | 0.96 | 2, 415 | .385 | .005 |

**Table 8. Means and standard deviations for perceptions of African identity based on historical conditions.**

| Perceptions of African identity | Positive | | Negative factual | | Negative colonial-perspective | |
|---|---|---|---|---|---|---|
| | *M* | *SD* | *M* | *SD* | *M* | *SD* |
| Self-definition | 5.24 | 1.27 | 5.03 | 1.37 | 5.03 | 1.31 |
| Self-investment | 5.81 | 0.92 | 5.68 | 1.06 | 5.71 | 0.98 |
| Collective self-esteem | 5.16 | 0.79 | 4.99 | 0.87 | 5.14 | 0.82 |
| Perceived descriptive group norms | 3.94 | 0.91 | 3.78 | 0.96 | 3.81 | 0.98 |
| Group entitativity | 5.17 | 0.91 | 5.13 | 1.02 | 5.11 | 1.00 |
| Identity motives | 4.37 | 1.84 | 4.36 | 1.96 | 4.13 | 2.02 |
| Desired possible future African identities | 6.05 | 1.71 | 5.74 | 1.94 | 6.01 | 1.79 |
| Feared possible future African identities | 2.68 | 2.05 | 2.91 | 2.13 | 2.59 | 1.93 |

## Discussion

Following from Study 1, we predicted in this study that positive historical representation compared to other historical representations would lead to more positive perceptions of African identity, which would, in turn, predict more collectively-oriented identity management strategies. As in Study 1, results provided no support for our predictions. Below, we consider implications of the findings, including methodological (and by extension theoretical) features of this study that inform the interpretation of the null findings obtained.

One consideration for interpreting the null effects is possible reactance to the baseline historical representation–that is, the colonialist narrative–that was presented in every condition. Psychological reactance occurs when individuals feel that the freedom to make behavioural choices are limited and threatened by a certain stimulus [49]. Consequently, individuals may adopt behaviours that oppose the stimulus as a means of restoring behavioural freedom [50]. In the present case, the baseline historical representation may have affected participants' responses because they perceived such a narrative of Africa as racist and objectionable, which subsequently affected their engagement and/or the degree to which they were willing to be influenced by the manipulations (i.e., the second historical representation video).

**Table 9. ANOVA results for the effect of historical condition on identity management strategies (IMS).**

| IMS | *F* | *df* | *p* | $\eta_p^2$ |
|---|---|---|---|---|
| Individual mobility | 0.09 | 2, 401 | .917 | .000 |
| Assimilation | 0.47 | 2, 401 | .623 | .002 |
| Individuation | 0.56 | 2, 401 | .563 | .003 |
| Social competition | 0.25 | 2, 401 | .782 | .001 |
| National subordinate re-categorisation | 0.27 | 2, 401 | .767 | .001 |
| Ethnic subordinate re-categorisation | 1.64 | 2, 401 | .195 | .008 |
| Revaluation comparison dimension | 0.09 | 2, 401 | .918 | .000 |
| New comparison dimension: education | 0.69 | 2, 401 | .503 | .003 |
| New comparison dimension: happiness | 0.54 | 2, 401 | .585 | .003 |
| New comparison dimension: culture | 1.56 | 2, 401 | .212 | .008 |
| New comparison dimension: community harmony | 1.03 | 2, 401 | .358 | .005 |
| New comparison dimension: religion | 1.29 | 2, 401 | .278 | .006 |
| New comparison dimension: morals | 1.15 | 2, 401 | .317 | .006 |
| New comparison dimension: family | 1.64 | 2, 401 | .195 | .008 |
| Collective political action | 0.50 | 2, 401 | .609 | .002 |

**Table 10. Means and standard deviations for identity management strategies (IMS) based on historical conditions.**

| IMS | Positive | | Negative factual | | Negative colonial-perspective | |
|---|---|---|---|---|---|---|
| | *M* | *SD* | *M* | *SD* | *M* | *SD* |
| Individual mobility | 3.06 | 1.69 | 3.13 | 1.66 | 3.06 | 1.65 |
| Assimilation | 3.20 | 1.27 | 3.36 | 1.51 | 3.34 | 1.34 |
| Individuation | 2.88 | 1.64 | 3.11 | 1.70 | 3.00 | 1.67 |
| Social competition | 5.61 | 1.16 | 5.51 | 1.41 | 5.54 | 1.27 |
| National subordinate re-categorisation | 0.00 | 1.35 | -0.13 | 1.42 | -0.03 | 1.42 |
| Ethnic subordinate re-categorisation | 0.52 | 1.67 | 0.15 | 1.79 | 0.20 | 1.71 |
| Re-evaluation of comparison dimension | 0.19 | 1.46 | 0.15 | 1.18 | 0.19 | 1.30 |
| New comparison dimension: education | 0.09 | 1.51 | 0.11 | 1.08 | 0.25 | 1.18 |
| New comparison dimension: happiness | 0.25 | 1.76 | 0.39 | 1.54 | 0.20 | 1.52 |
| New comparison dimension: culture | 0.63 | 1.57 | 0.64 | 1.51 | 0.34 | 1.75 |
| New comparison dimension: community harmony | 0.47 | 1.61 | 0.39 | 1.64 | 0.21 | 1.52 |
| New comparison dimension: religion | 0.19 | 2.03 | 0.45 | 1.58 | 0.13 | 1.67 |
| New comparison dimension: morals | 0.48 | 1.68 | 0.26 | 1.66 | 0.18 | 1.72 |
| New comparison dimension: family | 0.87 | 1.55 | 0.80 | 1.58 | 0.56 | 1.44 |
| Collective political action | 5.18 | 1.27 | 5.12 | 1.30 | 5.27 | 1.28 |

Reactance may have been facilitated by the assumption of an outgroup audience in the form of the research team and institution leading the research due to information presented on the consent form [51]. This information included the email addresses of the researchers which were United Kingdom-based and the Western names of the research supervisors and chair of the psychology ethics committee. This information could have raised doubts about the 'true' motives and purposes of the research. Indeed, some participants contacted the first author to raise concerns around the 'actual' purposes of the research and group membership of the authority behind the research.

The baseline historical representation coupled with the perceived outgroup status of the researchers may thus have been evocative enough to overwhelm any subsequent effect of the manipulation. Our justification for this methodological choice–of a baseline historical representation–was the notion that participants likely had some perception of African history and we posited this would typically be in line with the ubiquitous colonialist narrative because of how it has shaped the geopolitical and socio-cultural climate of Africa [31,32]. While these potential methodological problems may have undermined our ability to test the effect of different historical representations, the suggestion that these factors (i.e., potential reactance, audience effects) may play such a role is potentially an indicator of the power of historical representations. We now discuss other limitations and by implication, the overall research methodology (for Studies 1 & 2) before pointing out future research directions and theoretical implications.

## General discussion

The present research tested the effect of historical representations of the African people on perceptions of African identity and identity management strategies of Africans across two studies. In Study 1, we tested the effect of two historical representations of the African people: positive (prestigious precolonial African history and resistance to the colonial power) and negative (inhumane practices of precolonial Africans). In Study 2, we tested the effect of two historical representations of the African people: positive (prestigious precolonial African history) and negative factual (inhuman practices of precolonial Africans). We predicted that positive (vs. negative) historical representation would lead to more positive perceptions of African identity,

which in turn would predict more collectively-oriented identity management strategies. Altogether, results provided no support for these predictions.

In addition to the limitations already discussed, a further limitation of these studies is that we did not directly assess participants' perceptions of the history of Africa; rather we simply measured participants' perception of the positivity or negativity of the aspects of African history that they watched. In other words, the manipulation check in Studies 1 and 2 did not measure participants' overall appraisals of African history. A major goal of this research was to test the effect of Africans' perceptions of African history by introducing Africans to positive historical representations of Africa's precolonial eras in order to counter the negativity of the colonialist narrative. In this vein, we cannot claim to have successfully varied Africans appraisals (other than valance) of African history. This has broader theoretical ramifications too. For example, our manipulation check might not be able to differentiate the effect of historical representations of the achievements of precolonial Africans from other positive aspects of African identity (e.g., excellent food) on how Africans feel towards Africa because they both primarily induce positive dispositions towards Africa. Mapping out individuals' appraisals of their ingroup's history is important in understanding the relationship between history, perception of one's social identity and group behaviour. For example, Licata and colleagues [17] found that Africans who appraised colonialism as exploitative as opposed to developmental supported reparative action (in the form of compensation and apologies from European governments for past colonial exploitation) via increased national identification. This kind of appraisal goes beyond simple valance (good vs. bad history), and needs to be addressed in future research.

Another limitation was the short length of time with which participants had to engage with the historical representations. Specifically, participants' outlook on African history will have been cultivated over a long period of time (i.e., their lifetime) and most likely would have been facilitated by the educational system they passed through [13]. It might therefore be an unrealistic goal to change participants' appraisals of African history with a 3–5 minute video narrative. This notion is paralleled by research on prejudice reduction with evidence that one-off interventions do not produce improvements to explicit racial attitudes or long-term, implicit racial attitudes [52,53]. The general idea is that such high-stakes worldviews (such as the appraisal of the group's history and racial attitudes) will need more effort or a higher level of engagement from participants to produce attitudinal changes (e.g., [24]).

A related critique to the short length of time participants had to engage with historical representations may be that the null effects reflect an unrealistic expectation of changes to our outcome variables due to the short-term, one-off nature of our manipulations. However, we believe this is not the case because there is substantial evidence of perceptions of social identity (e.g., [54–58]) and identity management strategies (e.g., [59–67]) being affected by short-term or one-off manipulations. Therefore, it was entirely reasonable to expect (1) that the manipulations adopted in our research could produce discernible change in the outcome variables employed; and (2) that such potential changes would be explained by in-the-moment shifts in perceptions of African identity.

Irrespective of participants' engagement with the manipulations in regards to length of time/amount of information, the null effects obtained might also indicate that operationalising change to individuals' appraisals of their group's history solely as an intrapsychic/cognitive process rather than as a social process is ineffective (see [68]). Group members interacting and validating historical representations with each other might be more effective, reflecting the fact that history is a collective product [4,6,18,24,69].

The absence of a control condition in Study 1 and the presence of a baseline historical representation in Study 2 also meant that we lacked a true control condition with which to compare the effects of historical representations. Our approach in adopting a baseline

historical representation (i.e., the colonialist narrative of Africa) in Study 2 gave participants the opportunity of altering their views towards African history especially with the sharp contrast of agency and self-determination of Africans in the prestigious precolonial representation [11,16]. However, we did not have a pre-experiment (i.e., true control) estimations of how Africans perceived African identity and engaged in different identity management strategies. This is a limitation because the effectiveness of historical representations as an intervention could not be assessed due to the lack of a comparison with a true control condition. Theoretically, a true control comparison is important to any paradigm trying to change Africans appraisals of African history because of the consensual notion of a negatively evaluated African identity amongst Africans [2,12–14,31].

One further potential reason for our null findings could be that African identity (and thus, African history and futures) is simply not important or relevant to the sampled population. Contradicting this possibility, the mean levels of social identity-related variables (see Tables 3 & 8) such as in-group self-definition and self-investment [21] reveal that mean scores are above the mid-point of the scales, suggesting that African identity is a relevant and meaningful construct to Africans. A related implication of the null findings might be to question the applicability of SIT to an African context (bearing in mind that SIT was developed in a Western context): is it that there is no African identity or is it that the group processes are different in an African context? Indeed, Yuki [70] found that SIT predictions accounted well for group behaviours of North Americans but not for group behaviours of East Asians, because group behaviours in an East Asian context are guided by an intragroup focus on interpersonal networking among ingroup members and not achieving high ingroup status–that is, positive distinctiveness through intergroup comparison (also see Yuki and colleagues [71]). We cannot rule out the possibility that SIT may not account for group behaviours in an African context; hence, we remain open to the possibility that effects consistent with SIT may not be as apparent in an African context. At the same time, though, examination of the correlations between measures of perception of African identity and identity management strategies (see Tables 1 & 6) does suggest that relationships between variables are consistent with SIT.

### Future research directions

The findings, limitations, and (under-researched) context of this research present ample possibilities for future research. First, we suggest that the baseline historical representation from Study 2 should be dropped in favour of a true control condition without historical content because of potential reactance. Specifically, participants in the true control condition could watch a video unrelated to African history. This would facilitate a true test of the impact of historical (as opposed to other forms of) representations.

Second, future research should aim to reduce the potential concerns participants may have about an outgroup audience, which should allow for a less reactance-affected test of the current hypotheses. As an example, Eller, Koschate, and Gilson ([51]; Study 1) varied the audience that witnessed imaginary embarrassing incidents of participants by altering the group membership of the audience either to that of the ingroup or outgroups. They found that the level of embarrassment participants felt varied as a result of the group membership of the audience, thereby showing the potency of the audience to participants' responses (also see [72,73]). In order to buffer a potential outgroup audience effect in the present context, participants could receive information portraying pictorial evidence of the first author's (African) identity and ethnicity with a reiteration of the motives and purposes of the research to create a sense of an ingroup audience. Such a buffer should limit the potential impact that perceptions of the research team and institution have on participants responses.

Third, more detailed and nuanced assessment of how participants actually appraise ingroup (African) history would be valuable. For example, a key dimension of appraisals of history might be how relevant group history is in informing options to the solution of present-day challenges of the group. Such a measure would represent a better manipulation check, and could potentially be used to assess appraisals of collective history in other contexts. Perhaps surprisingly, no such measurement tool exists despite the conceptual importance placed on appraisals of an ingroup's past (e.g., [4,6,7,33,34,74–76]).

Fourth, responses to the treatments could conceivably differ between African subpopulations because appraisals of history may vary based on participants' more local social contexts. As such, future research could examine such potential differences. For example, responses to the treatments may vary between Native Africans (i.e., Africans living in Africa) and Africans living in the diaspora due to differences in baseline knowledge of African history that may be attributed to history curricula.

Fifth, in order to rigorously test the impact of historical representations, it may be beneficial for participants to watch full episodes of the history documentaries from which we sourced the short clips. For example, participants could be randomly assigned either to watch the full 6-part episodes of Africa's great civilisation documentary [16]; or watch 6 episodes of wildlife programme on Africa over a period of weeks. This would greatly strengthen the manipulation, addressing the ineffectiveness associated with one-off interventions in producing attitudinal change (e.g., [53]). Moreover, this would give participants an opportunity to effectively engage with their pre-treatment attitudes and personal dispositions towards African history, which may not be possible in short, one-off interventions [77]. Ultimately, such a manipulation would give a high level of elaboration of information and may engender systematic processing of information by participants which may increase attitudinal change [78,79].

Sixth, in order to capture the social processes (e.g., members of the ingroup deliberating issues) involved in changing worldviews/attitudes, future studies should examine whether knowing about or discussing others' views of historical representations validates and reinforces the effect of historical representation. For example, Morton and Duck [80] found that persuasive information on an anti-drug campaign was more effective in heightening adolescents' perceptions of risk indirectly through discussions with their parents who engaged with the anti-drug campaign also. This highlights the importance of communication with others in changing attitudes. In an experimental setting, participants can be given (factual) positive feedback on the effects of prestigious precolonial African history on changing other Africans' appraisals of African history. Such an indirect social process (pathway) to attitudinal change may be promising because it makes salient descriptive norms which are potent in determining group behaviour [23,24,81,82]. This corresponds with evidence that group-based discussions increase group identification, belief in the efficacy of collective action, and commitment to engage in collective action [68]. Accordingly, future research could adopt a group-based discussion (versus no discussion) paradigm whereby participants will have the opportunity to interact with other ingroup members after watching history documentaries on precolonial Africa.

## Conclusion

The research we have presented here does not provide support for the predicted role of historical representations in influencing perceptions of social identity and identity management strategies in the African context. We suggest that caution should be applied in interpreting these null findings as strong evidence against the hypotheses because of potential reactance to the baseline historical representation, potential outgroup audience effects, and the short-term nature (weakness) of our experimental manipulations. These factors may have limited the

effect of the manipulation in this research but this itself points towards various avenues for future research to help us better understand the role of historical representation in the African context.

## Author Contributions

**Conceptualization:** Damilola Makanju, Andrew G. Livingstone, Joseph Sweetman.

**Data curation:** Damilola Makanju.

**Investigation:** Damilola Makanju.

**Methodology:** Damilola Makanju, Andrew G. Livingstone, Joseph Sweetman.

**Supervision:** Andrew G. Livingstone, Joseph Sweetman.

**Writing – original draft:** Damilola Makanju.

**Writing – review & editing:** Damilola Makanju, Andrew G. Livingstone, Joseph Sweetman.

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
