## [Decision Letter · Decision Letter 0]

20 Jan 2020

PONE-D-19-26598

Testing the effect of historical representations on collective identity and action

PLOS ONE

Dear Mr Makanju,

Thank you for submitting your manuscript to PLOS ONE. After careful consideration, we feel that it has merit but does not fully meet PLOS ONE’s publication criteria as it currently stands. Therefore, we invite you to submit a revised version of the manuscript that addresses the points raised during the review process.

Both reviewers and myself found the theoretical perspective informing this study to be very well articulated. We also felt that the measures were well selected, and mapped well onto the theory. Both reviewers felt that this is the kind of paper with null results that is very good to have in the literature, and that it could serve as a springboard for others doing similar work. The reviews are at the end of this email. Reviewer 1 advocates for outright acceptance, and Reviewer 2 suggests a major revision.

Reviewer 2 makes some comments that would be worth addressing given the null results before the paper is ready for potential publication. Many of the potential reasons for the null results reported thus far are rather conceptual (e.g. reactance regarding an outgroup audience). It would be good to address Reviewer 2’s points about the potential methodological reasons for the null effects (e.g. the short time period in which the manipulation takes place). I know you have already given this some mention, and I think Reviewer 2’s comments give a very detailed account of how the paper could be improved in this regard.

When responding to their comments, I think it would be best to focus specifically on the third paragraph from the bottom as a guiding principle (I have pasted it below). The rest of the review can be used to interpret the context of the paragraph. I am not sure the conclusions need to be outright restructured per-se, unless you feel during your revision that it is the best strategy. I think it could also work well to simply add additional language responding to this point during the discussion, where it is alluded to already.

*“This implies that using an experimental methodology would entail considering what parts of identity might be capable of short-term response to a short-term and relatively minor stimulus as opposed to measures of longer-term identity adjustments that are probably less responsive to short term manipulation. This also implies the need for an experimental design with greater power, either a larger sample size or greater controls, perhaps (as the authors suggest) using a pretest-posttest comparison.**The authors’ discussion is consistent with what I am saying here about the nature of social representations and identities and the likelihood that Africans living in the UK would have well-developed strategies for responding to colonialist representations but, in my view, do not go far enough in thinking through the limitations of experimental methods for studying the phenomenon of interest, and giving serious consideration of what kinds of responses could be expected to arise in a short-term manipulation.”*

I think it would also be a good idea to assure that there are no other experimental studies examining your questions, or very similar ones, for context as Reviewer 2 notes as well.

We would appreciate receiving your revised manuscript by Mar 05 2020 11:59PM. To enhance the reproducibility of your results, we recommend that if applicable you deposit your laboratory protocols in protocols.io, where a protocol can be assigned its own identifier (DOI) such that it can be cited independently in the future. For instructions see: http://journals.plos.org/plosone/s/submission-guidelines#loc-laboratory-protocols

We look forward to receiving your revised manuscript.

Kind regards,

Geoffrey Wetherell, Ph.D.

Academic Editor

PLOS ONE

Journal Requirements:

Reviewers' comments:

Reviewer's Responses to Questions

**Comments to the Author**

1. Is the manuscript technically sound, and do the data support the conclusions?

Reviewer #1: Yes

Reviewer #2: Partly

2. Has the statistical analysis been performed appropriately and rigorously? 

Reviewer #1: Yes

Reviewer #2: Yes

3. Have the authors made all data underlying the findings in their manuscript fully available?

Reviewer #1: Yes

Reviewer #2: Yes

4. Is the manuscript presented in an intelligible fashion and written in standard English?

Reviewer #1: Yes

Reviewer #2: Yes

5. Review Comments to the Author

Reviewer #1: I thought this was a really interesting paper on an important topic. Though the findings were null, the authors did an excellent job clearly presenting them and offering reasonable next steps for continuing to explore options and mechanisms that might have produced their null results. Such a good job, in fact, that they addressed my most pressing concerns in their discussion. My one point left is that I think the actual conversational implementation of a social effect would work better than telling participants about others' reactions (48-49). But, really, I think this is exactly the sort of paper that should be published more often, as it is not only interesting in its own right, but also helps provide a springboard for other people working in the area.

Reviewer #2: This experimental study presents a well-developed theory and used carefully-designed measures and got null results across the board. There was no significant or substantive difference in respondents’ collective identities about Africa between those who had been presented with colonialist stereotypes of pre-colonial African history versus those who had been presented with either positive or negative but non-colonialist depictions of pre-colonial Africa.

The review of the literature and the development of hypotheses from the literature is well done. There is ample reason to believe from the literature that historical depictions do affect collective identities.

Scientifically, the question is whether the null results reflect on the theory or whether the null results are due to the operationalization of the concepts, the link between theory and experimental operationalization.

Do these results mean historical representations have no effect on identities? The authors do not think so, and neither do I. Both the authors and I believe they demonstrate that national/ethnic self-definitions are too strongly rooted to be affected by a few minutes of view of textual and visual materials. As the authors note, educational systems and international media have created a strong backdrop of colonialist depictions. Modern educated Africans, especially those taking web-based surveys and even more especially those living overseas (as a majority of these respondents are) seem likely to have strongly-developed self-perceptions that have either accepted or resisted these depictions.

It seems likely that the experimental manipulation is too short-term to have a big effect on a strongly-held identity. Returning to the literature, although I have not read the cited studies (except for #17 which I downloaded to check), I can find no evidence from the review of the literature or the summary in the article that any of the research on national identity and social representations involved experimental manipulations and instead examines the cross-sectional structure of correlations among constructs.

We are thus faced with evaluating the scientific importance of the null findings in the tests of hypotheses about whether a short-term manipulation of historical representations can have a non-negligible effect on people’s national identities and strategies of identity maintenance. If I may draw an analogy, there is substantial research that a substandard diet or stress effects health, but it is known that these effects are cumulative and you would not necessarily expect a short-term manipulation of stress or food consumption in a laboratory setting to have an effect on height or heart disease, although they might affect short-term bodily responses such as pulse rate or blood pressure or glucose levels.

This implies that using an experimental methodology would entail considering what parts of identity might be capable of short-term response to a short-term and relatively minor stimulus as opposed to measures of longer-term identity adjustments that are probably less responsive to short term manipulation. This also implies the need for an experimental design with greater power, either a larger sample size or greater controls, perhaps (as the authors suggest) using a pretest-posttest comparison.

The authors’ discussion is consistent with what I am saying here about the nature of social representations and identities and the likelihood that Africans living in the UK would have well-developed strategies for responding to colonialist representations but, in my view, do not go far enough in thinking through the limitations of experimental methods for studying the phenomenon of interest, and giving serious consideration of what kinds of responses could be expected to arise in a short-term manipulation.

My recommendation is that the study be published after revisions that (1) carefully scour the literature to determine whether there have been any experimental studies of the relations of interest and, if so, what the character of these manipulations and findings were and (2) modify the literature review and the conclusion to attend to the possible differences between the long-term effects of social representations that arise in correlational studies and the short-term effects that might be manipulated in an experimental context.

The sample size is small and cannot really support analyses of the effects on subpopulations, but the authors might consider a post hoc exploratory analysis to determine whether there are any suggestions of different responses to the treatment effects depending on where the respondents are living or what their nationality is (where Nigerian versus other may be the only available contrast).

6. PLOS authors have the option to publish the peer review history of their article (what does this mean?). If published, this will include your full peer review and any attached files.

Reviewer #1: No

Reviewer #2: No

---

## [Author Response · Author response to Decision Letter 0]

29 Feb 2020

Many thanks for your positive, encouraging response to the initial version of this paper, and we hope that you agree that acting on them has further improved the manuscript.

The revisions include (1) the addition of the critique and presentation of evidence on the plausibility of deriving short-term changes to our outcome variables in the General Discussion, and (2) the addition of the proposal of future research into possible differences between African subpopulations in the future research section of the General Discussion. 

As requested, we have responded to each comment and suggestion of the reviewers in bullet points below each comment/suggestion. We hope that this adequately addresses concerns raised regarding the previous version of the manuscript.

Editor’s Comments

Reviewer 2 makes some comments that would be worth addressing given the null results before the paper is ready for potential publication. Many of the potential reasons for the null results reported thus far are rather conceptual (e.g. reactance regarding an outgroup audience). It would be good to address Reviewer 2’s points about the potential methodological reasons for the null effects (e.g. the short time period in which the manipulation takes place). I know you have already given this some mention, and I think Reviewer 2’s comments give a very detailed account of how the paper could be improved in this regard. When responding to their comments, I think it would be best to focus specifically on the third paragraph from the bottom as a guiding principle ... The rest of the review can be used to interpret the context of the paragraph. I am not sure the conclusions need to be outright restructured per-se, unless you feel during your revision that it is the best strategy. I think it could also work well to simply add additional language responding to this point during the discussion, where it is alluded to already. 

• We thank you for the feedback on our manuscript and the outcome of the review. In accordance with your recommendation, we feel that the best strategy for our revision was to add additional points in the General Discussion to address the points made by Reviewer 2. 

I think it would also be a good idea to assure that there are no other experimental studies examining your questions, or very similar ones, for context as Reviewer 2 notes as well.

• To the best of our knowledge, our research is the first experimental inquiry into the effect of historical representations on perceptions of social identity and identity management strategies. Moreover, the research setting and sample (i.e., African identity) makes our investigation very novel in the field of social identity and intergroup relations. However, there are ample experimental studies that examine the effects of social factors (other than historical representations) on the perception of social identity and identity management strategies. We briefly review such studies in our response to Reviewer 2's comments. 

Reviewers’ Comments

Reviewer 1

I thought this was a really interesting paper on an important topic. Though the findings were null, the authors did an excellent job clearly presenting them and offering reasonable next steps for continuing to explore options and mechanisms that might have produced their null results. Such a good job, in fact, that they addressed my most pressing concerns in their discussion. My one point left is that I think the actual conversational implementation of a social effect would work better than telling participants about others' reactions (48-49). But, really, I think this is exactly the sort of paper that should be published more often, as it is not only interesting in its own right, but also helps provide a springboard for other people working in the area.

• We are very grateful for this positive feedback. In relation to your comment on your preference for a conversational implementation of a social effect (i.e., group discussions) over telling participants about others’ reactions, we feel this is an empirical question. This is because of the substantial evidence in the literature that the salience of descriptive norms is crucial in determining group behaviour. As such, telling ingroup members (i.e., Africans) about other ingroup members’ positive experiences in relations to learning about African history may be an important step to change Africans worldviews/attitudes towards the appraisals of African history. However, we do agree that group discussions would be a more engaging method, and definitely has potential to shape the (real-life) social processes that are involved in changing worldviews/attitudes towards the appraisals of African history. Ultimately, we hope that having these two possibilities as future research directions will encourage others to carry out such investigations. 

Reviewer 2

This experimental study presents a well-developed theory and used carefully-designed measures and got null results across the board. There was no significant or substantive difference in respondents’ collective identities about Africa between those who had been presented with colonialist stereotypes of pre-colonial African history versus those who had been presented with either positive or negative but non-colonialist depictions of pre-colonial Africa. The review of the literature and the development of hypotheses from the literature is well done. There is ample reason to believe from the literature that historical depictions do affect collective identities. Scientifically, the question is whether the null results reflect on the theory or whether the null results are due to the operationalization of the concepts, the link between theory and experimental operationalization. Do these results mean historical representations have no effect on identities? The authors do not think so, and neither do I. Both the authors and I believe they demonstrate that national/ethnic self-definitions are too strongly rooted to be affected by a few minutes of view of textual and visual materials. As the authors note, educational systems and international media have created a strong backdrop of colonialist depictions. Modern educated Africans, especially those taking web-based surveys and even more especially those living overseas (as a majority of these respondents are) seem likely to have strongly-developed self-perceptions that have either accepted or resisted these depictions. It seems likely that the experimental manipulation is too short-term to have a big effect on a strongly-held identity. Returning to the literature, although I have not read the cited studies (except for #17 which I downloaded to check), I can find no evidence from the review of the literature or the summary in the article that any of the research on national identity and social representations involved experimental manipulations and instead examines the cross-sectional structure of correlations among constructs. We are thus faced with evaluating the scientific importance of the null findings in the tests of hypotheses about whether a short-term manipulation of historical representations can have a non-negligible effect on people’s national identities and strategies of identity maintenance. If I may draw an analogy, there is substantial research that a substandard diet or stress effects health, but it is known that these effects are cumulative and you would not necessarily expect a short-term manipulation of stress or food consumption in a laboratory setting to have an effect on height or heart disease, although they might affect short-term bodily responses such as pulse rate or blood pressure or glucose levels. This implies that using an experimental methodology would entail considering what parts of identity might be capable of short-term response to a short-term and relatively minor stimulus as opposed to measures of longer-term identity adjustments that are probably less responsive to short term manipulation. This also implies the need for an experimental design with greater power, either a larger sample size or greater controls, perhaps (as the authors suggest) using a pretest-posttest comparison.

• We thank you for this positive feedback, and agree (in line with the conclusions made in our manuscript) that a strongly-held identity as African is shaped over the long term, and that to meaningfully shape the perceptions of African identity and identity management of Africans through historical representations needs more long-term, cumulative interventions.

The authors’ discussion is consistent with what I am saying here about the nature of social representations and identities and the likelihood that Africans living in the UK would have well-developed strategies for responding to colonialist representations but, in my view, do not go far enough in thinking through the limitations of experimental methods for studying the phenomenon of interest, and giving serious consideration of what kinds of responses could be expected to arise in a short-term manipulation. My recommendation is that the study be published after revisions that 

(1) Carefully scour the literature to determine whether there have been any experimental studies of the relations of interest and, if so, what the character of these manipulations and findings were and 

• To the best of our knowledge, our research is the first experimental inquiry into the effect of historical representations on perceptions of social identity and identity management strategies. Neveretheless, we believe that there is good reason to think that our outcome variables would show discernible effects of a short-term experimental manipulation. Specifically, we adopted the variables of perception of social identity and identity management strategies in our studies because there is substantial evidence of these sorts of variables being affected by short-term experimental manipulations of other social factors, albeit in identity contexts other than national identities. More importantly, the manipulations in previous research are relatively benign or subtle, and are often delivered in ways that are less powerful than the manipulations presented in our research. We have added a paragraph to the General Discussion on page 45 to summarise the requested literature review, and elaborate on the main points here.

In terms of the first step of our model, there is good evidence that experimental manipulations can produce discernible effects on perceptions of social identity (Jetten, Spears & Manstead, 1997b; Ellemers, Kortekaas & Ouwerkerk, 1999; Haslam, Oakes, Reynolds & Turner, 1999; Verkuyten & Hagendoorn, 1998; Doosje, Ellemers & Spears, 1995). As an example Jetten, Spears and Manstead (1997b) were able through a linguistic framing manipulation to induce higher levels of identification among students in relation to their academic discipline (psychology) in a high salience (social identity) condition in comparison to a low salience condition. Specifically, participants were made to feel positive about their social identity in the high salience condition and negative about their social identity in the low salience condition. This was achieved by having participants indicate their agreement or disagreement with extremely negative statements and moderately positive statements about their group in the high salience condition, and moderately negative statements and extremely positive statements about their group in the low salience condition. Consequently, identification with the group was increased in the high salience condition because participants endorsed more positive statements in comparison to the endorsement of more negative statements in the low salience condition. 

Similarly, Ellemers, Kortekaas and Ouwerkerk (1999) through a self-assignment manipulation were able to induce higher levels of identification (i.e., self-categorisation, commitment to the group and collective self-esteem) among minimal (i.e., ad hoc) groups in a high salience condition in comparison to a low salience condition. Specifically, participants were made to identify higher with their group if they self-assigned themselves to their group (in a high salience condition) and not when they were assigned to their group by the experimenter (in a low salience condition). Furthermore, Haslam, Oakes, Reynolds and Turner (1999) were able through a ‘three things’ manipulation to induce higher importance of national identity among Australians in a social identity salience condition in comparison to a personal identity salience condition. This was achieved by having participants write down and by extension reflect on three things they do often, rarely, well and badly either as a group member in the social identity salience condition or as an individual in the personal identity salience condition. Altogether, such findings illustrate that in-the-moment perception of a salient social identity can be impacted by experimental manipulations. 

Regarding the second step of our model, manipulations of social structural variables – that is, permeability of group boundaries, legitimacy of group status, stability of group status – have been shown to influence the identity management strategies employed by group members (Jackson, Sullivan, Harnish & Hodge, 1996; Ellemers, van Knippenberg, de Vries & Wilke, 1988; Ellemers, van Knippenberg & Wilke 1990; Taylor, Moghaddam, Gamble & Zellerer, 1987; Wright, Taylor & Moghaddam, 1990; Ellemers, Wilke and van Knippenberg, 1993; Commins & Lockwood, 1979b; Ellemers, 1993; Wright 1997). As an example, Jackson, Sullivan, Harnish and Hodge (1996) found that manipulations of permeability of group boundaries – the notion that changing group membership individually from one’s group to another is extremely difficult or impossible – among low-status social identities impacted the choice of identity management strategies employed by participants. Specifically, participants employed more social creativity strategies when they were made to believe that group boundaries were impermeable and not permeable through textual informational manipulations. 

Furthermore, Ellemers, van Knippenberg and Wilke (1990) found that manipulations of stability of status relationship – the notion that one’s group status position is extremely difficult or impossible to change – impacted the employment of social change strategies by group members. Specifically, group members were more willing to improve their group’s status collectively when they were made to believe that their group’s status position was unstable rather than stable through textual informational manipulations. Moreover, Ellemers, Wilke and van Knippenberg (1993; Experiment 1) found that manipulations of the legitimacy of status relationship – the notion that one’s group status position is a legitimate outcome of a just procedure – impacted the employment of collective identity management strategies. Specifically, group members of a low-status social identity were more willing to engage in collective behaviour to improve their group’s status when they were made to believe that their group’s status position was illegitimate and not legitimate, but only when the group’s status position was unstable rather than stable through textual informational manipulations. Additionally, Wright (1997; Experiment 1) found that a referential information manipulation, by a having confederate express anger (vs. absence of anger) at the ingroup’s low-status increased group members willingness to engage in collective active action. Altogether, such findings illustrate that identity management strategies can be influenced by short-term or one-off experimental manipulations. 

In conclusion, while we accept that our manipulations were not going to produce fundamental changes to peoples’ African identity, it was still entirely reasonable to expect (1) that the manipulations adopted in our research could produce discernible changes in the outcome variables employed; and (2) that such potential changes would be explained by in-the-moment shifts in perceptions of African identity. As such, we believe that the studies we report represent reasonable tests of our hypotheses due to the weight of evidence suggesting that the outcome variables employed in our research can be influenced by short-term or one-off experimental manipulations. 

(2) Modify the literature review and the conclusion to attend to the possible differences between the long-term effects of social representations that arise in correlational studies and the short-term effects that might be manipulated in an experimental context.

• We hope that our response above, on the experimental evidence of short-term or one-off manipulations influencing individuals’ perception of social identity and identity management strategies, has addressed this point, and made clearer why we believe that our experimental paradigm was a fair and reasonable test of our hypotheses given the nature of our outcome variables. In other words, we believe our manipulations were optimised to detect discernible changes to perception of African identity and identity management strategies of Africans, while also acknowledging that the effects of such manipulations are unlikely on their own to produce deeper, long-lasting change. More importantly, in accordance with the Editor’s recommendation, we have added additional sections to the General Discussion to address this point. 

The sample size is small and cannot really support analyses of the effects on subpopulations, but the authors might consider a post hoc exploratory analysis to determine whether there are any suggestions of different responses to the treatment effects depending on where the respondents are living or what their nationality is (where Nigerian versus other may be the only available contrast).

• While we are entirely open in principle to considering differences between subpopulations in terms of responses to the treatments, we did not have any specific, justifiable hypotheses regarding such potential differences. As such, we focused on establishing basic/generic effects across the sample as a whole. Moreover, as you have pointed out our current sample size is not adequate to have a robust test of such potential differences. These are possibilities that we will consider more closely in the future. More importantly, in order to encourage such a test and to reflect your review, we also now speculate briefly on these in the future research section of the General Discussion (p 48). Furthermore, we mention in the manuscript that our data are openly available on the project OSF site at https://osf.io/esnb4/?view_only=0be6351ee8b34ea8b9543afa101e5824, so that readers have an opportunity to investigate such (post-hoc) potential differences for themselves, and to carryout other exploratory analysis that may help to inform future research.

---

## [Editor Report · Decision Letter 1]

16 Mar 2020

Testing the effect of historical representations on collective identity and action

PONE-D-19-26598R1

Dear Dr. Makanju,

We are pleased to inform you that your manuscript has been judged scientifically suitable for publication and will be formally accepted for publication once it complies with all outstanding technical requirements.

With kind regards,

Geoffrey Wetherell, Ph.D.

Academic Editor

PLOS ONE
---

## [Editor Report · Acceptance letter]

23 Mar 2020

PONE-D-19-26598R1 

Testing the effect of historical representations on collective identity and action 

Dear Dr. Makanju:

I am pleased to inform you that your manuscript has been deemed suitable for publication in PLOS ONE. Congratulations! Your manuscript is now with our production department. 

With kind regards,

on behalf of

Dr. Geoffrey Wetherell 

Academic Editor

PLOS ONE